# NEMOTRON-CORTEXA: Enhancing LLM Agents for Software Engineering Tasks via Improved Localization and Solution Diversity

Atefeh Sohrabizadeh [* 1]   Jialin Song [* 1]   Mingjie Liu [1]   Rajarshi Roy [1]   Chankyu Lee [1]   Jonathan Raiman [1]
Bryan Catanzaro [1]

## Abstract

Large Language Models (LLMs) have demonstrated significant potential in code generation by following natural language instructions. Unfortunately, crucial real-world software engineering tasks, such as debugging or repository-level feature implementation, involve processing extensive contexts beyond current LLM context sizes and performing complex reasoning that is brittle using standard autoregressive decoding. Enhancing LLMs' performance in these scenarios requires careful consideration of the contextual information provided to the model, optimizing how the model leverages that, and identifying tools that enable more effective navigation of the development environment. To address these challenges, we introduce NEMOTRON-CORTEXA, an agentic system built on a predefined scaffold that enhances LLMs' ability to navigate and reason efficiently in complex software engineering contexts. Specifically, we develop a novel code embedding model that retrieves the most relevant files with greater precision, along with a localization agent that refines the granularity of the retrieval process. Additionally, we demonstrate that providing diverse contextual information and utilizing different prompt formats enable the model to identify and resolve issues more efficiently. We evaluate NEMOTRON-CORTEXA using SWE-bench (Jimenez et al., 2023), a benchmark derived from real-world GitHub issues. Compared to the widely used Agentless framework (Xia et al., 2024), NEMOTRON-CORTEXA achieves a higher issue resolution rate at a lower cost, highlighting its practical impact in addressing real-world software engineering challenges.

*Equal contribution   [1]NVIDIA, Santa Clara, CA, USA. Correspondence to: Atefeh Sohrabizadeh <asohrabizade@nvidia.com>, Jialin Song <jialins@nvidia.com>.

*Proceedings of the 42nd International Conference on Machine Learning*, Vancouver, Canada. PMLR 267, 2025. Copyright 2025 by the author(s).

## 1. Introduction

Recent advancements in Large Language Models (LLMs) have demonstrated remarkable capabilities across various domains, ranging from creative writing to analytical reasoning (Wei et al., 2022; Yuan et al., 2022; Gómez-Rodríguez & Williams, 2023; Ahn et al., 2024). The influence of LLMs on software engineering has been especially widespread and recognized. Early models showcased their proficiency in generating syntactically correct code and solving interview-style programming questions (Chen et al., 2021; Roziere et al., 2023; Li et al., 2023; Guo et al., 2024). Building on this progress, recent efforts in LLM research on programming have shifted from snippet and function-level generation towards developing LLM *coding agents* capable of tackling entire real-world software engineering tasks (Yang et al., 2024; Xia et al., 2024; Wang et al., 2024a; Antoniades et al., 2024; Liu et al., 2024b; Xie et al., 2025). These *coding agents* are equipped to utilize tools commonly used by human engineers, including Command Line Interfaces (CLI) and Integrated Development Environments (IDE), to perform critical tasks such as debugging and applying patches in a repository. A key benchmark for evaluating the effectiveness of these LLM agents is SWE-bench (Jimenez et al., 2023), which features real-world software engineering problems derived from GitHub issues, aimed at generating patches that successfully resolve them. The rapid improvements on the SWE-bench leaderboard highlight the growing interest in building powerful LLM software agents and underscore their potential to automate software engineering tasks. In this work, we propose several ways to simultaneously enhance the efficiency and accuracy of LLM agents within this context, and use SWE-bench as a benchmark.

System design decisions play a crucial role in the end-to-end performance of an LLM software agent, since real-world software engineering tasks often involve processing a large code repository and reasoning about relations among modules. A practical example of the impact of system design can be found in the *coding agents* of OpenHands (Wang et al., 2024a) and SWE-Agent (Yang et al., 2024): the authors allow an agent to perform free-form action sequences guided by system prompts that outline the available action

types. While such free-form flows can be effective, the large action space often results in many costly iterations and makes debugging difficult. To mitigate this, Agentless (Xia et al., 2024) employs a more structured approach, guiding the LLM through a predetermined sequence of steps. Despite these advances, challenges remain to accurately locate the problem and generate correct repairs.

We present NEMOTRON-CORTEXA (*C*ode *O*ptimization for *R*epository *T*asks with *EX*ecution *A*gents), a coding agent designed to strengthen issue localization and repair generation. First, we develop a dedicated code embedding model that optimizes the retrieval of relevant files and a localization agent that refines the granularity of issue localization. Second, we equip the localization agent with programming tools based on Abstract Syntax Tree (AST) and Language Server Protocol (LSP) that enable compact code navigation by taking advantage of multi-step reasoning. Third, we adopt an ensemble of LLMs approach that improves localization accuracy and helps identify additional relevant code elements that can improve the repair process. The inspiration for ensembling was drawn from our observation that localization agents instantiated with different LLMs perform particularly well on different problems, as also noted in (Zhang et al., 2024a). We can explain this benefit by observing that even when models fail to accurately identify the exact entity (function or class) causing an issue, they nonetheless often identify closely related entities. Our optimizations here improve the localization precision over Agentless, a competitive approach on SWE-bench that is widely used in model benchmarking (Liu et al., 2024a; Jaech et al., 2024), by $18.08\%$ and recall by $11.32\%$, on average. In addition, we show that the introduction of diversity in the repair stage can further enhance performance. To induce diversity, we develop a simple and efficient method that leverages different contextual information and multiple prompt formats during patch generation. This approach outperforms standard temperature sampling (Brown et al., 2024; Xia et al., 2024) and allows NEMOTRON-CORTEXA to achieve a higher issue resolution rate while minimizing the number of inference calls for patch generation by $4.4\times$ compared to Agentless. NEMOTRON-CORTEXA outperforms Agentless by achieving an issue resolution rate of 42.00% on the SWE-bench Lite and 52.60% on SWE-bench Verified, while costing only $0.51 per instance.

In summary, our main contributions are as follows:

- We develop a code embedding model specialized in retrieving relevant code chunks to a given bug, achieving state-of-the-art file retrieval accuracies on the SWE-bench benchmark.

- We design a localization agent that integrates advanced programming tools and leverages an ensemble of LLMs to deliver more precise and granular issue localization.

- We propose a diverse solution generation method that leverages different contextual information and varied prompt formats, significantly enhancing sample efficiency.

- Experimental results demonstrate that NEMOTRON-CORTEXA outperforms Agentless, OpenAI's go-to approach for showcasing their real-world coding performance, while being more cost-effective.

## 2. Related Works

### 2.1. LLMs for Code Generation

LLM progress in code generation has been driven by research efforts to specialize them through continued pretraining on code data (Guo et al., 2024; Bai et al., 2023; Roziere et al., 2023; DeepSeek-AI et al., 2024) sourced from open repositories (Kocetkov et al., 2022; Lozhkov et al., 2024) and commit histories (Muennighoff et al., 2023). Further enhancements come from reinforcement learning (Le et al., 2022) and, more commonly, instruction fine-tuning. Instruction fine-tuning incorporates techniques to solve complex coding challenges (Luo et al., 2024), broadens model capabilities by leveraging unlabeled open source code (Wei et al., 2023; Yu et al., 2024; Wu et al., 2024), ensures solution accuracy with self-generated tests (Chen et al., 2022), and improves code validation and debugging via interactions with LLM agents (Lei et al., 2024).

### 2.2. LLM Agents for Software Engineering

Real-world software engineering tasks are more complex than solving interview questions as evidenced by the development of SWE-bench (Jimenez et al., 2023). Recent rapid progress in designing LLM agents highlights the importance of this application. SWE-agent (Yang et al., 2024) enables an LLM agent to interact with the development environment for tasks like search and edit. AutoCodeRover (Zhang et al., 2024b) builds a more sophisticated code search tool based on the abstract syntax tree. Multi-agent systems perform a divide-and-conquer strategy by developing specialized agents for each sub-task (Liu et al., 2024b; Wang et al., 2024a). Agentless (Xia et al., 2024) takes a different approach by designing a multi-step pipeline where LLMs complete each step. Iterative approaches and self-reflections, such as in (Chen et al., 2023; Shinn et al., 2024) improve coding performance, with SWE-Search (Antoniades et al., 2024) integrating self-feedback within a Monte-Carlo Tree Search framework to refine strategies.

### 2.3. Code Retrieval

Code retrieval is a critical task in domains like code explanation, code summarization, and documentation lookup, as

shown by recent benchmarks (Li et al., 2024; Wang et al., 2024b). Early efforts to create code-specific embedding models include BERT-based (Devlin, 2018) models such as CodeBERT (Feng et al., 2020) and UniXcoder (Guo et al., 2022). While there has been a growing interest in developing decoder-only LLM-based embedding models for text retrieval tasks (Neelakantan et al., 2022; Moreira et al., 2024b; Meng et al., 2024; Lee et al., 2024), few models target code retrieval. Notable exceptions include the models developed by Voyage AI (Voyage AI, 2024) and OpenAI (Neelakantan et al., 2022). More recently, CodeXEmbed (Liu et al., 2024c) emerged as a family of code embedding models, excelling in many code retrieval tasks. However, limited focus has been given to code retrieval for identifying buggy code based on issue descriptions.

## 3. NEMOTRON-CORTEXA

### 3.1. Overview of the Pipeline

Since an entire codebase frequently exceeds the context window of present LLMs, we are obligated to subsample the relevant files to provide a context that fits yet is sufficient for resolving the issue. Consider that issue descriptions are typically under 1000 tokens, while repository codebases contain orders of magnitude more tokens. For example, SWE-bench instances have on average 195 words per issue for 438k lines of code (Jimenez et al., 2023). Our approach is therefore to use the issue description to retrieve relevant files that form a context that fits and is likely to be sufficiently informative to resolve the issue. This retrieval not only ensures the model has the right context but also helps filter out irrelevant or confusing information. Therefore, similar to Agentless (Xia et al., 2024), the NEMOTRON-CORTEXA pipeline is divided into two main stages: *localization* and *repair*, as shown in Figure 1. The localization stage involves two steps: first, identifying the most relevant files (Section 3.2), and then refining the granularity of the retrieval to focus on specific functions, classes, or methods (Section 3.3). During the repair stage, we combine different contexts and prompts to increase patch diversity (Section 3.4), followed by a filtering process to select the most promising patch (Section 3.5).

### 3.2. File Localization with Code Embedding Model

Accurately localizing the issue to the relevant files is essential for the success of the pipeline. If this step fails, the model cannot produce an effective patch, making the remainder of the pipeline irrelevant. To tackle this, Agentless creates a concise repository representation, similar to the Linux `tree` command, and provides it to the LLM to identify the top N suspicious files. It also experiments with using a general embedding model but finds that the embedding-based approach performs worse than the prompt-based method. However, the prompt-based approach has its own limita-

tions: the LLM is expected to make decisions based solely on file names, which can lower its accuracy, or depend on prior knowledge of these repositories. On the other hand, processing the entire codebase and passing the full contents of each file to it would be prohibitively expensive. To tackle this, we develop a coding embedding model specifically trained to retrieve the most relevant files based on the issue description. Our model, called NV-EmbedCode[1], is fine-tuned from an existing text-based embedding model that uses bidirectional attention followed by average pooling (Muennighoff et al., 2024; Lee et al., 2024; Moreira et al., 2024a).

The retrieval task involves identifying the "oracle" file — the file requiring edits to resolve the issue — based on the issue description. Since files can be large, we chunk them and prefix each chunk with meta-information, such as the full file path, to help the model maintain context about the chunk's location. We curate a dataset specifically for this purpose, mapping issues to code edits. The relevant passages for retrieval are the chunks from the oracle files. Once a chunk is retrieved, the oracle file is located by referencing its meta-information. The issue descriptions often may have redundant information, such as package version details, or insufficient details, like ambiguous buggy code with little explanation. To mitigate this problem, we prompt an LLM to generate a concise summary that includes all relevant information, such as filenames, functions, error messages, stack traces, and other technical details. We then create a second dataset containing these summaries, while selecting the positive passages the same way as before.

To enhance the embedding model's understanding of code data, we augment our training set with additional publicly available datasets for text-to-code (retrieving code documents based on a textual query), code-to-code (retrieving code documents based on a code query), and hybrid code (retrieving a combination of code and text documents based on a hybrid query) retrieval tasks, as categorized by (Li et al., 2024). We also extend the hybrid code category by generating a synthetic dataset where an LLM creates code in response to a query. To help the model differentiate between these datasets, we prefix the queries with dataset-specific instructions following the approach in prior works (Wang et al., 2023; Lee et al., 2024) with the template *"Instruct: {dataset_instruction}\n Query: {query}"*. Similar to NV-Embed (Lee et al., 2024), we mask out the instruction tokens during average pooling.

We train the embedding model with a contrastive learning objective (Gao et al., 2021) to maximize the similarity between the query and its relevant (positive) passage, while

---

[1]Available at https://build.nvidia.com/nvidia/nv-embedcode-7b-v1

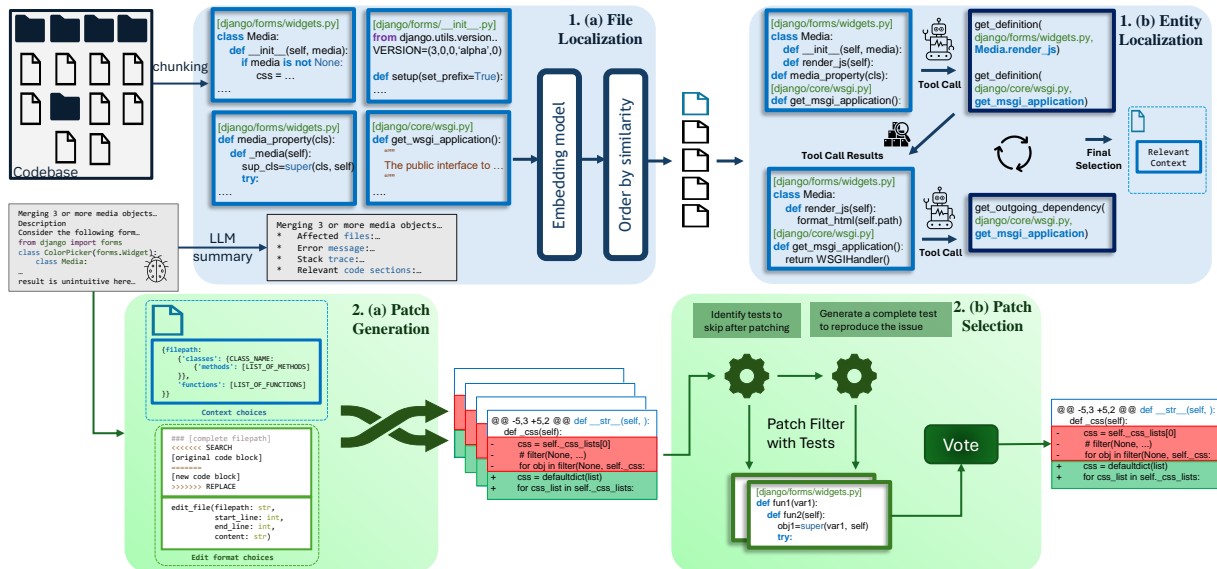

*Figure 1.* Overview of NEMOTRON-CORTEXA. From the issue description, NEMOTRON-CORTEXA first retrieves relevant files using NV-EmbedCode, our specialized code embedding model, then refines issue locations to granular entities with a localization agent. We use a novel strategy to generate diverse candidate patches and select one based on LLM-generated tests and majority voting.

minimizing the similarity to irrelevant (negative) passages. We adopt the positive-aware hard-negative mining technique from NV-Retriever (Moreira et al., 2024b) to minimize the likelihood of including false negatives in the data. Since the base model may not fully understand code data, we fine-tune it and subsequently mine new hard negatives from the refined model to improve the quality of the hard negatives. Each batch includes samples from all datasets, but we avoid using in-batch negatives to prevent conflicts between positive and negative samples. This problem occurs when chunking the oracle files to create positive passages, as a single query may correspond to multiple positive passages. If these query-positive pairs appear in the same batch, it can lead to misleading negative and positive samples.

### 3.3. Entity Localization with Localization Agent

With the code embedding model, we can obtain a short list of files relevant to the issue. However, the full texts of these files often still exceed a model's context window. We need to narrow down the issue locations further. A code file consists of high-level *entities*, such as classes and functions, so the next step of localization is to identify candidate entities from a few top-ranked files. Agentless uses a direct prompting approach, providing a skeleton format of each file containing *only* entity headers and asking an LLM to select the relevant ones. However, without access to the code content, the model lacks sufficient context to respond accurately. Inspired by information foraging theory in software debugging (Lawrance et al., 2010), we develop a localization agent that iteratively explores the repository by performing common actions such as looking up definitions

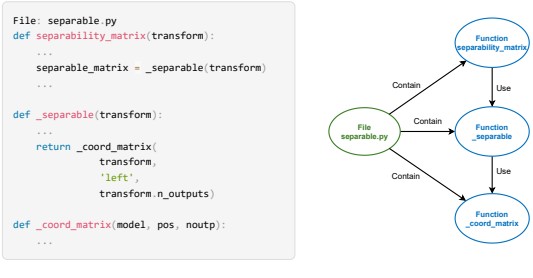

(a) Code Example      (b) Code Graph

*Figure 2.* Example code and its graph representation.

and tracing call stacks. To facilitate structured navigation of the repository, we start by building its graph representation. The nodes represent code files and entities – functions, classes, and class methods – extracted using AST. Each node stores the corresponding code text and its position metadata. We establish two types of directed edges: *contain* and *use*. The *contain* edges establish hierarchical relationships, connecting files to their functions and classes, and classes to their methods, thereby providing agents with a structural view of the repository. Complementarily, *use* edges capture functional dependencies, including function invocations and class instantiations. For example, in Figure 2, the graph contains a node for the file `separable.py` and a node each for the three functions `separability_matrix`, `_separable`, and `_coord_matrix`. The file node has an edge pointing to each function with the *contain* type. `separability_matrix` node has an edge pointing to `_separable` with the *use* type, as does `_separable` to `_coord_matrix`.

We equip the localization agent with three tools to navigate

through the graph. They are reminiscent of features in popular IDEs such as *Go To Definition* and *Call Hierarchy*. We use an LSP server to implement these tools.

- **get_definition(file_name, entity_name)**: returns the code definition of the specified entity in the requested file.

- **get_incoming_dependency(file_name, entity_name)**: returns the definition codes of the specified entity's source nodes using the list of its incoming edges.

- **get_outgoing_dependency(file_name, entity_name)**: returns the definition codes of the specified entity's destination nodes using the list of its outgoing edges.

Concurrent to our work, MarsCode Agent (Liu et al., 2024b) also builds a code knowledge graph from a repository with AST and LSP. Our version differs by focusing only on high-level entities and providing additional tools built on top of standard LSP functions.

The localization agent leverages the file ranking results from the code embedding model, focusing on the top-ranked ones. For these files, we create a concise representation containing only definition signatures of high-level entities. These signatures, combined with the issue description, are then fed into an LLM to identify a few entities relevant to the issue description. This step is similar to Agentless's direct prompting approach after identifying suspicious files (Section 3.1.2 in (Xia et al., 2024)). Next, the localization agent examines the full code of the chosen entities and utilizes tools to inquire about related entities. Each tool call returns a collection of new code snippets that the agent stores in its context for future queries. The agent iterates through multiple interactions with the repository to gather sufficient context. Since excessive tool calls can lead to context overload, the agent is tasked to filter the context and keep only the relevant ones in a memory buffer. After a few iterations, the localization agent is prompted to select a final set of entities requiring modifications to fix the issue.

### 3.4. Diverse Patch Generations

After localizing the issue and narrowing it down to specific suspicious entities, the next step is to generate a patch that resolves it. In this step, the LLM is prompted to generate a patch based on the provided code context. Instead of producing the entire code, the LLM typically is prompted to generate only the edit, either as a search/replace format (Xia et al., 2024; Aide, 2024) or a git diff format (Jimenez et al., 2023; Yang et al., 2024). Our observations indicate that the choice of context and prompt format significantly impacts the correctness of the generated patches, leading to varied sets of resolved instances. Specifically, LLMs are highly sensitive to the format of the requested edits, often yielding different solutions when only the patch format is changed,

even with all other factors held constant.

To explore this further, we studied the effect of two specific edit formats on the resolved instances of SWE-bench. These formats are illustrated in Figure 1. The first format, referred to as the `search/replace` format, consists of a *search* component that includes the original code snippet to be replaced and a *replace* component that specifies the replacement content. The second one is an `edit_file` API that specifies the file path, the starting and ending line numbers of the code to be edited, and the content to be inserted or modified. Additionally, we evaluated the effect of two different contexts within the code section of the prompt: the entities identified in Section 3.3 and the top-retrieved file from Section 3.2.

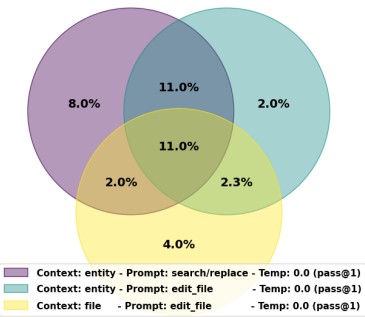

*Figure 3.* Venn diagram showing the distribution of the percentage of resolved instances across context and edit format changes for SWE-bench Lite. Different combinations of context and edit formats demonstrate varying strengths, here ranging from 19.3% to 32%, and cover different instances which, can contribute to a higher overall pass rate of 40.3%.

Figure 3 depicts the Venn diagram of the resolved instances when we vary these choices of contexts and edit formats. The numbers in each section indicate the percentage of resolved instances in SWE-bench Lite that fall into that subset. While it is expected that changes in context would result in different instances being resolved, the diagram also highlights that the final fix is highly sensitive to the choice of the edit format. Different combinations of context and edit format exhibit varying strengths in resolving the instances and cover very different instances. Existing approaches focus on identifying the single most effective choice and then rely on temperature sampling to generate diverse solutions. In contrast, we propose changing both the context and the edit format, in addition to temperature sampling. This method increases solution diversity, which in turn increases the number of resolved issues while reducing the number of inference calls needed for patch generation.

### 3.5. Patch Selection

After generating a diverse set of patches, we apply a series of filtering steps to identify the most promising patch as the

final solution. First, we remove solutions that fail to produce valid edit instructions or result in syntax errors. Next, we normalize the edited codes by removing comments, docstrings, and empty lines, as well as standardizing variable, function, and class names. Using these normalized versions, we record the frequency of the repeated solutions. We then use regression tests and the generated reproduction tests to filter out solutions that fail these tests. Our approach closely follows the methodology outlined in Agentless (Xia et al., 2024) and directly leverages their released artifacts[2]. In this process, Agentless first executes all existing tests in the repository to identify a subset of passing tests that successfully run on the original codebase. The passing tests are then analyzed by an LLM, which identifies any test that is not suitable to verify whether the issue has been correctly fixed. The remaining tests are designated as regression tests. For reproduction tests, Agentless generates a complete testing file designed to both reproduce the original issue described and verify whether the issue has been successfully resolved. These tests are executed on the original repository to filter out any tests that fail to detect the issue. We retain all valid reproduction tests at this stage. In cases where no solutions remain after filtering, we re-include all solutions available before applying the test. Finally, we apply majority voting on the remaining solutions based on the frequency of their repetition to select the final patch.

## 4. Experiments

### 4.1. Experiment Setup

To train the embedding model, we use a Parameter-Efficient Fine-Tuning (PEFT) technique called Low-Rank Adaptation (LoRA) (Hu et al., 2021). Specifically, we fine-tune a text embedding model with contrastive loss, applying LoRA with a rank of 16, alpha of 32, and a dropout rate of 0.1. Our base model is NV-EmbedQA-Mistral-7B-v2 (NV-EmbedQA-v2 for short) (Moreira et al., 2024a), a commercially available version of NV-Embed-v2 (Lee et al., 2024). We employ the Adam optimizer (Kingma, 2014) with 200 warm-up steps, a learning rate of 1e-6, and linear decay. The training is conducted using Bfloat16, with a maximum sequence length of 512 tokens. The model is fine-tuned with a batch size of 64, where each batch consists of a query, one positive document, 7 hard negatives, and we do not use in-batch negatives, as explained in Section 3.2.

The training data is curated from the SWE-bench training set (Jimenez et al., 2023) and a portion of the CoIR training set (Li et al., 2024), which includes data from APPS (Hendrycks et al., 2021), CoSQA (Huang et al., 2021), CodeTransOcean (Yan et al., 2023), StackOverflowQA (Overflow, 2021), and SyntheticText2SQL (Meyer

*Table 1.* Overall comparison over SWE-bench Lite and Verified. Claude refers to the Claude-3.5-Sonnet model.

| Method | LLM | % Resolved | | Avg. |
| | | Lite | Verified | $ Cost |
|---|---|---|---|---|
| OpenHands (Wang et al., 2024a) | Claude | 125 (41.67%) | **265** **(53.00%)** | – |
| | | 78 (26.00%) | – | 1.10 |
| Agentless (Xia et al., 2024) | Claude | 122 (40.67%) | 254 (50.80%) | – |
| | GPT-4o | 96 (32.00%) | – | 0.70 |
| Moatless Tools (Team, 2024a) | Claude | 115 (38.33%) | – | – |
| AutoCodeRover (Zhang et al., 2024b) | GPT-4o | 92 (30.67%) | 192 (38.40%) | – |
| SWE-agent (Yang et al., 2024) | Claude | 69 (23.00%) | 168 (33.60%) | – |
| **NEMOTRON-CORTEXA** | Mix | **126** **(42.00%)** | 263 (52.60%) | 0.51 |

et al., 2024). Additionally, we synthetically generate a dataset of code produced from coding queries in Code-Feedback (Zheng et al., 2024) by prompting DeepSeek-v2.5 (DeepSeek-AI, 2024). The LLM summaries mentioned in Section 3.2 are generated using Llama-3.1 405B (Dubey et al., 2024). The final dataset contains approximately 534k pairs of query-positive documents, as detailed in Appendix A.1

We build the code graphs using tree-sitter[3] to extract entities from code files. We implement the three code navigation tools in Section 3.3 with pylsp[4]. The localization agent starts with the top-6 files according to our code embedding model and performs a maximal of 5 rounds of entity localization with tools. The agent can issue multiple tool calls at each round. We instantiate the localization agent with four different LLMs: DeepSeek-v3 (Liu et al., 2024a), GPT-4o (OpenAI, 2024), Llama-3.3 70B (Dubey et al., 2024), and Qwen2.5-72B-Instruct (Team, 2024b; Bai et al., 2023). The final localization result is an ensemble of them. During localization, we use greedy sampling to generate responses. The patches are generated using Claude 3.5 Sonnet.

### 4.2. End-to-end Issue Resolution Results

We validate NEMOTRON-CORTEXA on SWE-bench (Jimenez et al., 2023) Lite and Verified, which consist of 300 and 500 instances, respectively. Table 1 summarizes the comparison to five representative open-source baselines: OpenHands (Wang et al., 2024a), Moatless Tools (Team, 2024a), AutoCodeRover (Zhang et al., 2024b), SWE-agent (Yang et al., 2024), and Agentless (Xia et al., 2024). NEMOTRON-CORTEXA

---

[2]https://github.com/OpenAutoCoder/Agentless/releases/tag/v1.5.0

[3]https://tree-sitter.github.io/tree-sitter/
[4]https://github.com/python-lsp/python-lsp-server

achieves the highest resolution rate of 42% (126 / 300) on Lite and the second highest rate 52.60% (263 / 500) on Verified. Notably, NEMOTRON-CORTEXA achieves its performance at an average cost of only \$0.51 per issue. In comparison, OpenHands reports a 26% resolution rate on SWE-bench Lite at \$1.10/issue, and Agentless reports 32% at \$0.70/issue.[5] Further scaling test-time compute in NEMOTRON-CORTEXA can yield higher resolution rates here. Our contributions drive NEMOTRON-CORTEXA's strong performances at minimal cost: our code embedding model and localization agent provide more accurate and informative contexts for LLMs, while our novel diverse solution generation strategy significantly reduces the number of candidate generations needed. We will break down their impacts in the next sections.

## 4.3. Embedding Model Analysis

We rely on the golden solutions provided in SWE-bench as the ground truth for identifying the oracle files. While it is possible to address the issue by editing other parts of the codebase, this ground truth offers a reliable base for comparing different approaches. We evaluate three approaches for this localization task: lexical-based (i.e., BM25), prompt-based, and embedding-based methods, as detailed in Section 3.2. The prompt-based results are taken from Agentless logs. For the embedding-based approach, we chunk the files to 450 tokens with no overlap. This choice stems from our experiment with existing embedding models, where we tested token lengths ranging from 450 to 4096 and overlaps of 0 or half the context window, and saw that the models perform better with smaller context windows (see Appendix A.1). This may be attributed to two factors: first, these models are often trained on short documents, and second, a smaller window allows the model to attend more to individual code chunks and focus on their semantics rather than normalizing the vector embedding across the entire file. Each chunk is further prefixed with the complete file path as additional metadata.

Table 2 presents a comparison of these methods on SWE-bench Lite and Verified sets. We evaluate NV-EmbedCode against state-of-the-art models, including SFR-Embedding-2_R (Meng et al., 2024), NV-Embed-v2 (Lee et al., 2024), and NV-EmbedQA-v2 (Moreira et al., 2024a). The evaluation metric is $recall@k$, where $k$ is the number of oracle files for a given instance. This metric measures the proportion of the oracle files correctly identified within the top $k$ results. The results indicate that existing embedding models struggle to perform well with the LLM summaries

[5]The costs for more recent, higher-performing submissions have not been reported. However, they are likely substantially higher as OpenHands now allows more inference calls per issue and Agentless uses Claude 3.5 Sonnet, which is priced higher than GPT-4o.

*Table 2. $Recall@k$ in retrieval for SWE-bench Lite and Verified, where $k$ is the number of oracle files, using either the original issue description or its LLM-generated summary as input.*

| Method | Model / Approach | Issue | Lite | Verified |
|---|---|---|---|---|
| Lexical | BM25 | Original | 42.33% | 40.67% |
| | | Summary | 38.00% | 41.75% |
| Prompt* | GPT-4o | Original | 63.00% | 65.55% |
| Embedding | SFR-Embedding-2_R | Original | 60.33% | 61.00% |
| | | Summary | 55.33% | 55.46% |
| | NV-Embed-v2 | Original | 56.00% | 59.90% |
| | | Summary | 57.67% | 59.62% |
| | NV-EmbedQA-v2 | Original | 61.33% | 62.81% |
| | | Summary | 55.67% | 59.20% |
| | **NV-EmbedCode** | Original | 67.67% | 68.37% |
| | | Summary | **70.33%** | **71.95%** |

\* Refers to Agentless results

and prefer the original descriptions. This is likely due to differences in linguistic styles and vocabulary of these summaries compared to the natural queries the models were originally trained on. The decrease in the performance of BM25 with these summaries for SWE-bench Lite suggests a reduction in exact term matches and the frequency of keywords. By developing our code-specific embedding model, NV-EmbedCode achieves significant improvements. Not only does it outperform existing embedding models with original issue descriptions, but it also achieves significantly better results with LLM-generated summaries. These summaries enhance the issue descriptions by eliminating redundancy and removing irrelevant details, which can improve their effectiveness for retrieval. The improvement in NV-EmbedCode is due to the inclusion of diverse code and synthetic data in our training set and the use of instruction templates, which enable the model to recognize and adapt to different linguistic styles. Although the prompt-based method can outperform embedding-based ones when paired with existing embedding models, it falls short compared to our model.

## 4.4. Entity Localization Analysis

Accurate entity localization provides focused context, increasing the likelihood of an LLM generating correct patches. NEMOTRON-CORTEXA employs localization agents to identify relevant entities. At the end of an agentic search trajectory, we combine its first iteration and last iteration responses to extract entities that balance the breadth and depth of localization. We refer to the first iteration results as Direct Prompting (DP), and the last iteration results as Localization Agent (LA). We compare our entity localization accuracy to Agentless (Xia et al., 2024), using results extracted from their logs, as their method includes a dedicated entity localization step. We use the golden solutions in

*Table 3.* Precision and recall metrics in retrieving oracle entities for SWE-bench Lite and Verified. DP denotes direct prompting entities and LA refers to localization agent results.

| Method | Lite | | Verified | |
|---|---|---|---|---|
| | Precision | Recall | Precision | Recall |
| Agentless | 15.90% | 59.94% | 17.54% | 58.37% |
| NEMOTRON-CORTEXA (DP) | 35.83% | 60.56% | 34.16% | 57.46% |
| NEMOTRON-CORTEXA (LA) | **38.91%** | 71.17% | **39.62%** | 64.04% |
| NEMOTRON-CORTEXA (LA+DP) | 35.84% | **74.72%** | 34.51% | **67.62%** |

*Table 4.* Localization agent's precision and recall metrics in retrieving oracle entities for SWE-bench Lite and Verified set when instantiated with different models. Ensemble of all four agents yields substantial gains in the recall.

| Model | Lite | | Verified | |
|---|---|---|---|---|
| | Precision | Recall | Precision | Recall |
| Qwen2.5-72B | 34.40% | 40.33% | 36.58% | 41.71% |
| Llama3.3-70B | 33.44% | 40.40% | 36.21% | 39.74% |
| Deepseek-v3 | 35.30% | 45.94% | **40.39%** | 46.94% |
| GPT-4o | 34.25% | 44.61% | 36.67% | 45.20% |
| Ensemble of Above | **35.84%** | **74.72%** | 34.51% | **67.62%** |

SWE-bench to extract ground truth entities (oracle entities).

Table 3 shows mean precision and recall results for Agentless and NEMOTRON-CORTEXA, together with its two components. Overall, NEMOTRON-CORTEXA achieves higher precision and recall than Agentless on both SWE-bench Lite and Verified. Its localization stage also costs less: $0.11 per instance for NEMOTRON-CORTEXA compared to $0.15 per instance for Agentless. The superior LA results demonstrate the advantage of combining specialized tools with multi-step reasoning. Adding DP results further boosts the recall by capturing relevant entities that might be overlooked during the agent's later, more focused iterations. Additionally, the dual approach of DP and LA also provides diverse high-quality contexts for repair generation and enables more efficient repair generations, which we discuss in the next section. NEMOTRON-CORTEXA employs an ensemble of four localization agents, each instantiated with a different LLM. Table 4 compares the ensemble results with those of the individual agent. The recall increases by 28.78% and 20.68% over the best individuals for the Lite and Verified sets, respectively, thus retrieving significantly more relevant entities. The precision also increases for the Lite set because the ensemble benefit from models' strengths in solving different problem subsets. This result points to an alternative axis for inference-time scaling by employing multiple models.

### 4.5. Repair Generation

As discussed in Section 3.4, changing the context and edit format leads to different instances being resolved. One approach here is to identify the best-performing combination of context and edit format and then generate multiple solutions using temperature sampling. For example, Agentless generates 40 patches per instance by sampling 4 edit locations and creating 10 patches in the `search/replace` format for each (1 using greedy sampling and 9 using temperature sampling). They report that the number of resolved instances increases until 40 patches but plateaus beyond that point.

Our observations indicate that we can achieve similar or better performance with fewer patches by increasing the diversity in how these patches are generated. Figure 4 illustrates the number of passed instances (out of 300 total) for SWE-bench Lite across different contexts, edit formats, and temperature settings. We consider three contexts here: 1) File, the top-1 file retrieved by NV-EmbedCode, 2) LA, and 3) DP. With greedy sampling (temperature 0.0), the LA context, and the `search/replace` format, we can already achieve a high pass rate of 96 instances, highlighting the effectiveness of our localization step. Adding a patch generated via temperature sampling increases the number of correctly resolved instances to 107 (pass@2). Interestingly, while these two strategies have the highest pass rates individually, their combination does not introduce many new resolved instances compared to other alternatives. For example, changing the context to File and the prompt format to `edit_file` results in 115 resolved instances overall, even though this new combination individually has the lowest pass rate. This pattern persists across other contexts and edit format combinations as well (Section 4.7), indicating that varying these factors increases solution diversity, ultimately leading to a higher total number of correct solutions while reducing the number of required inference calls.

While temperature sampling is less effective than changing other factors, it remains a useful strategy for generating new patches. In our final patch generation process, we consider 4 contexts (File, LA, DP, and LA+DP, where LA and DP outputs are merged), two edit formats (`search/replace` and `edit_file`), and two temperature settings (0.0 for greedy sampling and 0.8 for temperature sampling). We first generate 4 solutions with greedy sampling by using the File and LA as context, and both our edit formats. We then generate a single additional solution by greedy sampling the LA+DP as context and `edit_file` prompt. Finally, we generate 4 more solutions via temperature sampling, covering all four contexts, with two using the `search/replace` format and two using the `edit_file` format. These 9 solutions can achieve pass@1 performance (after applying the filtering steps as

*Table 5.* Precision and recall metrics in retrieving oracle entities for SWE-bench Verified. The localization agent starts with the top-6 files retrieved by different embedding models.

| Embedding Model | Precision | Recall |
|---|---|---|
| NV-EmbedQA-v2 (LA+DP) | 27.62% | 65.03% |
| NV-EmbedCode (LA+DP) | 34.51% | 67.62% |

detailed in Appendix A.3) of 126 for the Lite set and 263 for the Verified set, surpassing Agentless results of 122 and 254 with 40 patches. This demonstrates how diversifying the solutions can lead to a higher number of correct patches.

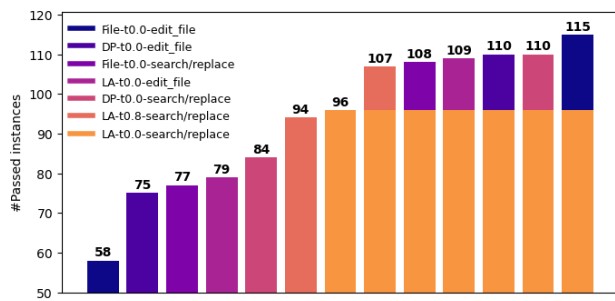

*Figure 4.* #Passed instances of SWE-bench Lite under different contexts – File, Localization Agent (LA), and Direct Prompting (DP) – with varying edit format and temperature settings. Two-segment bars show pass@2 when both input solutions are used.

### 4.6. Ablation on Localization

In this section, we provide ablation studies on SWE-bench Verified for our two contributions to localization: 1) our fine-tuned NEMOTRON-CORTEXA embedding model; 2) the localization agent for entity localization. First, Table 5 demonstrates that as NV-EmbedCode achieves higher $recall@k$ as shown in Table 2, the localization agent produces more accurate entity localization results. This improvement indicates that starting with more accurate, relevant files enhances the agent's success in identifying oracle entities.

Second, to study the impact of entity localization on final resolution, we compare using entity results (LA+DP) and the top-1 retrieved file (File) as context for patch generation. We generate 9 patches with both edit formats and temperature sampling and apply the filtering steps described in Section 3.5. As shown in Table 6, using entity localization results as contexts significantly improves the resolution rate, highlighting the importance of entity localization.

### 4.7. Ablation on Solution Diversity

In Section 4.5, we demonstrated that diversifying the solution generation strategies can increase the resolution rate. To further investigate it, we conducted an additional experiment in which, instead of the 9 diversified patches as described in Section 4.5, we generated patches using LA

*Table 6.* Pass@9 and Pass@1 results across different generation combinations for SWE-bench Verified with 500 instances. The primary diversification method is temperature sampling. We further evaluate the impact of introducing diversity by varying the edit format as described in Section 3.4 and using the four contexts mentioned in Section 4.5.

| Context | Edit Format | Pass@9 | Pass@1 |
|---|---|---|---|
| File, LA, DP, LA+DP | Both | 307 | 263 |
| LA | `search/replace` | 270 | 237 |
| LA | Both | 284 | 245 |
| DP | Both | 261 | 220 |
| LA+DP | Both | 286 | 239 |
| File | Both | 257 | 223 |

entities as context and the `search/replace` edit format– as it has shown to be the best single policy–while applying temperature sampling for diversity. We then extended this experiment by employing both edit formats while keeping the context fixed, and repeated the process across all four context types discussed in Section 4.5. Table 6 presents a comparison of the pass@9 and pass@1 (after applying the filtering steps outlined in Section 3.5) metric for each combination. These results reinforce our claim that diversifying both the edit format and context increases the number of correct patches, as each configuration brings distinct strengths and weaknesses. By combining them, we are able to leverage their complementary advantages.

## 5. Conclusion & Future Work

NEMOTRON-CORTEXA demonstrates that improved localization and diverse solutions are effective avenues to enhance LLM software agents for real-world tasks. Our comprehensive analyses validate three key contributions: 1) our new code embedding delivers state-of-the-art file retrieval accuracies here; 2) our localization agent enables precise issue location identification; 3) our diverse solution generation method, leveraging different contexts and edit formats, exhibits superior sample efficiency. NEMOTRON-CORTEXA outperforms Agentless on SWE-bench at a lower cost, showcasing its effectiveness.

We recognize that NEMOTRON-CORTEXA's resolution can be further improved by refining the selection process and/or scaling the number of generated patches. Currently, majority voting in the patch selection process is suboptimal, especially since we diversify the solutions generated, and a correct solution may not appear multiple times. In fact, by using LLM-based voting instead of majority voting, generating more accurate reproduction tests, and employing reasoning models during patch generation, NEMOTRON-CORTEXA's resolution on the Verified set can reach 68.4% while costing $3.28/issue. These optimizations are complementary to our approach, and a detailed analysis of the impact of each component is left for future work.

## Acknowledgments

The authors would like to acknowledge the constructive feedback from anonymous reviewers and the work of ICML 2025 program chairs and area chairs.

## Impact Statement

Reliable and capable LLM software agents have the potential to automate routine programming tasks, such as debugging, and boost developer efficiency by freeing them to work on more innovative projects and explore more efficient solutions. By providing natural language interfaces to programming tasks, these agents can also enable people without prior programming experience to use code in their work. This potential can obviously be misused to do harm by malicious actors, for example, to generate malware or automate cyber attacks. Thus it is important to build robust safety mechanisms in LLM agents to minimize their negative impacts. By carefully considering these factors, we can work to maximize the benefits of LLM software agents.

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

## A. Appendix.

### A.1. NV-EmbedCode Training Details

We employ three types of instructions for the various retrieval datasets used in training NV-EmbedCode:

1. **Bug-to-File**: Applied to the dataset derived from the SWE-bench training set, which maps issue descriptions to their respective "oracle" files - files requiring edits to resolve the issue. These oracle files are identified based on the pull request (PR) that resolved the issue successfully.

2. **BugSummary-to-File**: Also based on the SWE-bench training set, except that here the original issue descriptions are replaced with summaries generated by an LLM - Llama-3.1 405B.

3. **General**: Used for all other retrieval datasets, which include a diverse set of tasks, including text-to-code, code-to-text, code-to-code, and hybrid retrieval.

Table 7 provides an overview of these instruction templates along with the number of samples for each. Each training sample consists of one query-positive document pair and 7 hard negative documents.

*Table 7.* Instruction template and number of samples used for each category of NV-EmbedCode's training dataset.

| Dataset Type | Instruction Template | Number of Samples |
|---|---|---|
| Bug-to-File | Given a bug description, retrieve codes that need to be edited to resolve it | 177k |
| BugSummary-to-File | Given a summary of bug description generated by an LLM, retrieve codes that need to be edited to resolve it | 92k |
| General | Retrieve code or text based on user query | 265k |

The SWE-bench training set consists of 19,008 issues extracted from pull requests across 35 non-test repositories. We filter out instances in which the corresponding golden solution creates a new file or modifies non-Python files, resulting in a curated subset of 13,922 issues. For each remaining instance, the oracle files are designated as positive documents. As described in Section 3.2, all documents are segmented into chunks; in this setting, we use a chunk size of 512 tokens. We also include a portion of the CoIR training set (Li et al., 2024), which includes data from APPS (Hendrycks et al., 2021), CoSQA (Huang et al., 2021), CodeTransOcean (Yan et al., 2023), StackOverflowQA (Overflow, 2021), and SyntheticText2SQL (Meyer et al., 2024). Additionally, we synthetically generate a dataset of code produced from coding queries in CodeFeedback (Zheng et al., 2024) by prompting DeepSeek-v2.5 (DeepSeek-AI, 2024). Table 8 summarizes the number of samples for each dataset.

*Table 8.* Number of samples for each training dataset of NV-EmbedCode.

| Dataset Name | Main Retrieval Task | Dataset Type | Number of Samples |
|---|---|---|---|
| SWE-bench Train Set (Original issue) | Issue-to-Oracle-File | Bug-to-File | 177k |
| SWE-bench Train Set (LLM summary as issue) | Issue-to-Oracle-File | BugSummary-to-File | 92k |
| APPS (Hendrycks et al., 2021) | Text-to-Code | General | 5k |
| CoSQA (Huang et al., 2021) | Text-to-Code | General | 20k |
| SyntheticText2SQL (Meyer et al., 2024) | Text-to-Code | General | 100k |
| CodeTransOcean (Yan et al., 2023) | Code-to-Code | General | 1k |
| StackOverflowQA (Overflow, 2021) | Hybrid Code | General | 14k |
| CodeFeedback (Zheng et al., 2024) + DeepSeek-v2.5 (DeepSeek-AI, 2024) | Hybrid Code | General | 125k |

As described in Section 4.3, we evaluated a range of chunk sizes from 450 to 4096 tokens using the SFR-Embedding-Mistral model to determine the optimal chunking of code files. Our experiments on the SWE-bench Lite set revealed that a chunk size of 450 tokens yields the highest retrieval accuracy, as illustrated in Figure 5. Based on these findings, we adopted a chunk size of 512 tokens for training and 450 tokens for inference with our model. Consistent with the behavior observed in the SFR-Embedding-Mistral model, NV-EmbedCode also exhibits a decline in retrieval accuracy as chunk size increases, as shown in Figure 5. This can be attributed to two factors: first, a more localized view of the files enables the model to more effectively determine if the code snippet is related to the issue; second, larger chunk sizes will be out-of-distribution, as both our current training set and the training data used for the base model consist primarily of shorter documents.

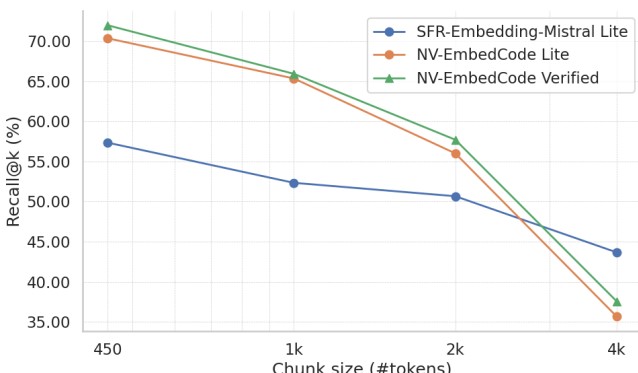

*Figure 5.* Recall@k in file retrieval for SWE-bench when changing the maximum number of tokens per each chunk. *k* is the number of oracle files based on the golden solution that resolved the issue.

*Table 9.* NEMOTRON-CORTEXA file retrieval accuracy compared to free-form agentic retrieval baselines for SWE-bench Verified. As baselines retrieve an arbitrary number of files, we report recall@$\infty$. For NEMOTRON-CORTEXA, we use recall@10, since other methods retrieved at least 15 files on average.

| Method | Recall@$\infty$ |
|---|---|
| OpenHands (Wang et al., 2024a) | 85.60% |
| AutoCodeRover-v2 (Zhang et al., 2024b) | 88.20% |
| MarsCode (Liu et al., 2024b) | 87.20% |
| SWE-agent-v1 (Yang et al., 2024) | 76.80% |
| **NEMOTRON-CORTEXA** | **94.00%** (recall@10) |

### A.2. Additional Results for Localization

**Compare with agentic file retrieval methods.** We compare file retrieval accuracy of NV-EmbedCode against four representative free-form agentic approaches: OpenHands (Wang et al., 2024a), AutoCodeRover-v2 (Zhang et al., 2024b), MarsCode Agent (Liu et al., 2024b), and SWE-agent-v1 (Yang et al., 2024). For each agent, we extract files from their official trajectory logs in the SWE-bench leaderboard. Since these methods can retrieve an arbitrary number of files, we denote the accuracy metric with recall@$\infty$. In this comparison, we use recall@10 for NEMOTRON-CORTEXA as other baseline methods retrieved at least 15 files on average. Table 9 shows that our code embedding model, NV-EmbedCode, enables NEMOTRON-CORTEXA to achieve significantly higher file recall accuracy compared to agentic baselines, demonstrating the effectiveness of finetuning for the specialized task of file retrieval from issue descriptions.

**Entity localization with different ensemble methods.** In Section 4.4, we discussed how NEMOTRON-CORTEXA utilizes an ensemble of four localization agents, each instantiated with a different model. Here, we explore two alternative ensemble strategies. The first uses temperature sampling with a single model to generate diverse outputs. The second aggregates entity results across all steps from a single agent. We use DeepSeek-v3 as it is individually the strongest in entity localization. For the temperature sampling ensemble, we run entity localization four times; once with temperature 0 and three times with temperature 0.8. When aggregating all steps, we use greedy sampling (temperature 0). Table 10 demonstrates that the model-based ensemble achieves higher entity recall than both alternatives, while the temperature sampling baseline obtains a higher precision by retrieving fewer entities. These results highlight that model diversity helps improve the recall accuracy of entity localization the most among the three ensemble methods.

**Entity Localization Accuracy and Resolution Status.** To further assess the impact of localization on the final resolution rate, we divide the instances into two sets, resolved and unresolved, based on whether any of the nine patches successfully resolve the issue. We then calculate the entity localization accuracy for each set. As shown in Table 11, we observe that resolved cases exhibit consistently higher precision and recall in localization compared to unresolved ones, indicating a strong correlation between localization accuracy and successful patch generation.

*Table 10.* NEMOTRON-CORTEXA's entity ensemble achieves higher recall accuracy compared to other ensemble baselines.

| Method | Precision | Recall |
|---|---|---|
| Temperature Sampling | **49.93%** | 57.03% |
| Aggregating Results of All Agent Iterations | 35.47% | 53.47% |
| Ensemble of Models (LA + DP) | 34.51% | **67.62%** |

*Table 11.* Average entity precision and recall accuracy as broken down by resolution results. Resolved instances have higher accuracies.

| Result | Lite | | Verified | |
|---|---|---|---|---|
| | Precision | Recall | Precision | Recall |
| Resolved | 44.54% | 86.87% | 46.05% | 90.91% |
| Unresolved | 27.58% | 63.20% | 32.75% | 64.06% |

## A.3. Solution Filtering

After generating 9 patches for each instance as described in Section 4.5, we have the correct patch for 146 instances of SWE-bench Lite and 307 instances of SWE-bench Verified. However, the PASS_to_PASS and FAIL_to_PASS tests provided by SWE-bench cannot be used to identify the correct patches. Table 12 summarizes the average number of patches remaining after each filtering step described in Section 3.5, as well as the number of resolved instances achieved by applying majority voting at each step (pass@1). The results demonstrate the effectiveness of these filtering steps but also indicate areas for improvement, which we leave for future work. Specifically, the regression and reproduction tests, generated using an LLM, may have inaccuracies that could result in removing correct solutions or failing to prune the wrong solutions. Furthermore, majority voting is not an ideal heuristic here, as certain incorrect solutions may repeat multiple times, whereas some correct solutions may only be generated once or fewer times than the incorrect ones. Additionally, in cases of ties between solutions based on their frequency, we choose one of them randomly. A more robust scoring mechanism can help mitigate these limitations.

*Table 12.* The impact of filtering techniques on the average number of remaining patches and the number of resolved instances when majority voting is applied after filtering (pass@1).

| Filtering | Lite | | Verified | |
|---|---|---|---|---|
| | avg. | % Resolved | avg. | % Resolved |
| Syntax check | 7.7 | - | 7.9 | - |
| + Normalization | 5.7 | 107 (35.67%) | 5.5 | 225 (45.00%) |
| + Regression test | 4.6 | 116 (38.67%) | 4.5 | 242 (48.40%) |
| + Reproduction test | 4.0 | 126 (42.00%) | 4.0 | 263 (52.60%) |

## A.4. Cost Analysis

For measuring costs, we use the following prices for the models we employ:

- Claude 3.5 Sonnet API: $3/M input tokens and $15/M output tokens.

- GPT-4o API: $2.5/M input tokens and $10/M output tokens.

- DeepSeek-V3: $0.9/M input tokens and $1.1/M output tokens.

- Llama 3.3-70B: $0.59/M input tokens and $0.73/M output tokens.

- Qwen2.5-72B: $0.4/M input tokens and $0.75/M output tokens.

- NV-EmbedCode: As this is a Mistral 7B-based model, we use Mistral pricing of $0.11/M input tokens. We cache the chunks and query our vector database for repeated ones.

Table 13 summarizes a breakdown of the cost for each step of our pipeline. SWE-bench Lite and Verified have a total of 800 instances, but 93 instances are common between them. We consider the common ones only once in the cost calculation.

*Table 13.* Breakdown of the cost for running NEMOTRON-CORTEXA on the 707 unique instances of SWE-bench Lite and Verified, considering the 93 instances common between them only once.

| Stage | Model | #Input Tokens M | #Output Tokens M | Total Cost USD (%) | Avg. Cost / Instance USD (%) |
|---|---|---|---|---|---|
| File Localization | NV-EmbedCode | 217.95 | – | 23.97 (6.69%) | 0.03 (6.69%) |
| Entity Localization | Mix | 42.66 | 1.74 | 53.68 (15.00%) | 0.08 (15.00%) |
| Generating Tests | GPT-4o | 13.00 | 5.74 | 89.90 (25.11%) | 0.13 (25.11%) |
| Patch Generation | Claude-3.5-Sonnet | 47.63 | 3.17 | 190.44 (53.20%) | 0.27 (53.20%) |
| Total | Mix | 321.24 | 10.65 | 358.00 (100%) | 0.51 (100%) |

