# OpenReview forum: "Nemotron-CORTEXA: Enhancing LLM Agents for Software Engineering Tasks via Improved Localization and Solution Diversity"
_ICML.cc/2025/Conference — ICML 2025 poster_

### Official Review · Reviewer_sboJ · 2025-03-08

**Overall Recommendation:** 3

**Summary:**

This paper proposes CORTEXA, a coding agent for bug-fixing based on Agentless v1.5. CORTEXA differs from Agentless in two key ways: 1) it uses an embedding-based file-retrieval (using a finetuned embedding model) combined with an agentic entity retrieval approach instead of direct prompting for localization, 2) it increases patch diversity by combining different temperatures, contexts, and prompt formats when sampling repair patches. Combined this yields slightly better performance at (probably) lower cost.

### Update After Rebuttal
I thank the authors for their detailed response and would like to encourage them to include the additional ablation results in a revised version for the paper. These additional results convince me of the effectiveness of the proposed approach beyond Agentless and agentic retrieval approaches. I have updated my score accordingly.

**Claims And Evidence:**

Key claims:
* Their embedding-based file retrieval approach outperforms a purely prompt-based approach => Well supported in Table 2, although additional ablations would strengthen this result.
* Their agentic retrieval approach combining different models and retrieval stages improves entity retrieval recall over Agentless => Well supported in Tables 3 (when combined with ensembling across models) although interesting baselines such as using the same model at temperature or combining different retrieval stages from an individual model.
* Their diversity-focused patch generation approach performs better than simply sampling at temperature => Well supported for pass@2 and two samples in Figure 4, with no ablations for the full end-to-end fixes with 9 generations.
* Combining these components leads to a notably stronger method than Agentless => limited support with actual performance difference to the underlying Agentless being minimal <2% and no statistical significance reported (t-test for SWE-Lite gives only p=34%)

**Essential References Not Discussed:**

No critical works missing, although a broader overview of current code agents could be provided, see e.g. the 26 different agents compared to in Agentless.

**Experimental Designs Or Analyses:**

I checked the soundness of all experiments reported in the main paper and they seem consistent and reasonable. However, more baselines and other methods could significantly strengthen the conclusions:
In Table 2, an additional embedding baseline of generating descriptions of to-be-retrieved code (and similar for the embedded code) and direct agentic retrieval like SWE-Agent or AutoCodeRover.
In Table 3, a comparison to other agentic retrieval approaches (e.g. AutoCodeRover v2, or even SWE-Agent v1).
It should perhaps be made more explicit when CORTEXA (and Agentless) use an ensemble (e.g. Table 2 (Prompt only) and Table 3).

**Methods And Evaluation Criteria:**

The dataset choice (SWE-Bench), evaluation criteria, and conducted ablation studies are mostly appropriate. Comparisons based on cost with unclear API prices for cited reference are hard to evaluate. Comparing at equal number of pre-filtering patches with Agentless may have yielded statistically significant improvements.

**Other Comments Or Suggestions:**

* I would suggest a table format with fewer (especially vertical) lines to improve readability.
* Computing p-values using paired sample tests could strengthen claims.

**Other Strengths And Weaknesses:**

### Strengths
* Focus on efficiency via diversity in ensembling and as a result matching/improving performance at a lower cost is an important and interesting direction
* Demonstrates the effectiveness of FineTuning embedding models specifically for code retrieval in the bug-fixing setting
* Confirms the effectiveness of agentic entity retrieval and combining results across retrieval stages
* Shows the effectiveness of increasing diversity before patch aggregation using different edit formats and contexts.

### Weaknesses
* In multiple places relevant baselines are not considered, limiting conclusions to be relative to Agentless.
* Important details are not described (e.g. how is the context filtered (Section 3.3)? What prompts are used?)
* Without these details and the trained embedding model (or more details on its training process), the results are not reproducible.
* Variability and statistical significance of results are not measured/discussed but p-values computed with the available data show no statistically significant improvement over Agentless, despite a significantly more complex approach.

**Questions For Authors:**

1) Will the finetuned model, the generated datasets, or the agent traces be released?
2) Ho was the context filtered by the code agent (Section 3.3)
3) How does localization cost compare to Agentless? How would an Agentless approach perform at a matched token cost in Table 3?
4) Can you discuss the ablation results on different chunking strategies mentioned around line 335 (left)?
5) Can you compare to other localization methods in Tables 2 and 3 (see above)?

**Relation To Broader Scientific Literature:**

Comparison to other methods in Table 1 is quite limited with many approaches being relatively old and e.g. the closely related MarsCode Agent (published September 2024) not being included. A more detailed comparison of retrieval performance to e.g. MarsCode Agent, and AutoCodeRover (using a similar agentic approach and published in April 2024) should be included. While the authors claim MarsCodeAgent to be contemporary it was released more than 4 months before this submission.

**Theoretical Claims:**

None

---

> ### Author Rebuttal · Authors · 2025-04-01
>
> We truly appreciate the time and effort you have put into reviewing our work and are grateful for your insightful comments that help improve our paper. For comparisons to SOTA works, please refer to our response to Reviewer WZSG. We would like to highlight that Cortexa outperforms Agentless in several key areas:
> 1. Patch Generation Efficiency: Our reported performance is with only 9 patches, whereas Agentless requires 40 patches. When generating a comparable number of patches as mentioned in response to Reviewer WZSG (32 patches), Cortexa significantly outperforms Agentless, achieving 58.8% on the Verified set (p-value = 0.006).
> 2. Localization Performance: While Agentless uses a combination of prompt-based and embedding-based approaches, our specialized embedding model achieves higher retrieval accuracies. From Table 2 in the Agentless paper, its localization cost is `$0.15/instance`, while Cortexa’s localization cost is `$0.11/instance` (including file and entity localization) as shown in Appendix A.2.
> 3. Recall in Oracle Entity Identification in Table 3: Our recall rate in identifying the oracle entity is statistically superior to Agentless, with a p-value of 0.0007.
>
> We will release our code including all the prompts we used, full logs, and our embedding model upon the paper’s publication.
>
> **Additional baselines on retrieval:** Here are the new baseline results on the Verified set:
>
> **1. Temperature sampling and ensembling across multiple stages instead of ensembling multiple models:** We use DeepSeek-v3 here as it is individually the strongest in entity localization. Neither method can match Cortexa’s recall. The temperature sampling approach obtains a higher precision since it retrieves fewer entities. However, recall matters more for patch generation to provide contextual information. The results show that model diversity is the most effective approach here.
> |Model|Precision|Recall|
> |-|-|-|
> |Temp Sampling (deepseek-v3)|49.93%|57.03%|
> |All stages (deepseek-v3)|35.47%|53.47%|
> |Cortexa (DP+LA)|36.22%|68.09%|
>
> **2. Agentic retrieval baselines:** We select 4 agentic approaches here: OpenHands, AutoCodeRover-v2, MarsCode Agent, and SWE-agent-v1. For each agent we extract files from their official trajectory logs in the SWE-bench leaderboard. Since these methods can retrieve an arbitrary number of files, we denote the accuracy metric with recall@inf. In this comparison, we use recall@10 for Cortexa as other baseline methods retrieved at least 15 files on average.
>
> For entity retrieval accuracy, these agentic approaches often do not explicitly retrieve entities. Moreover, there is no exact way to measure entity retrieval results from their logs as there is no fixed pattern. Thanks to its localization agent step, Cortexa provides a concrete way to measure the more granular entity retrieval accuracy, which proved helpful to iterate and improve the agent.
> |Model|Recall@inf|
> |-|-|
> |OpenHands|85.6%|
> |AutoCodeRover-v2|88.2%|
> |MarsCode|87.2%|
> |SWE-agent-v1|76.8%|
> |Cortexa|94.0% (recall@10)|
>
> **Impact of diversifying generations:** To address this comment, we run the following experiments on the Verified set:
> 1. Taking the context (localization agent, LA) and the edit format (search/replace) that give us the highest pass rate with temperature 0 and generating 9 patches for it (1 greedy sampling and 8 temperature sampling).
> 2. Changing the edit format to use both edit formats.
> 3. Changing the context to other choices of DP (direct prompt), LA+DP, and File that uses the results of the embedding model.
>
> Table below summarizes the pass@9 and pass@1 (after applying the filtering steps outlined in the paper) results for each combination. These results support our claim that diversifying both the edit format and context increases the number of correct patches, as each combination has its own strengths and weaknesses, and diversification allows us to leverage all of them.
> |Context |Edit Format|pass@9|pass@1|
> |-|-|-|-|
> |All|Both|307|263|
> |LA|Search/replace|270|237|
> |LA|Both|284|245|
> |DP|Both|261|220|
> |LA+DP|Both|286|239|
> |File|Both|257|223|
>
> **Context filtering:** In our prompts, we ask the agent to filter irrelevant context by issuing a special `<keep>` command after consuming a list of code context. We will provide these prompts upon code release.
>
> **Chunking strategy:** We experimented with the SFR-Embedding-Mistral model starting with a chunk size of 4k and gradually reducing it. We observed a recall@k of 43.67% at a chunk size of 4k on the Lite set and it peaked at a chunk size of 450 with recall@k of 57.33%. Based on these results, we selected a chunk size of 450 for both training our model and final inference. Similar to the SFR-Embedding-Mistral model, our model's retrieval accuracy decreases with larger chunk sizes. For instance, with a chunk size of 4k, the retrieval accuracy on the Lite set is 35.67%, compared to 70.33% with a chunk size of 450. We will include the full experimental results in the paper.

---

### Official Review · Reviewer_KBKm · 2025-03-14

**Overall Recommendation:** 2

**Summary:**

- This paper introduces CORTEXA, a software agent that involves training a model specifically for localizing the right files, building a localization agent to identify the right entities within a file, and a workflow for diverse patch generation and selection.
- CORTEXA outperforms Agentless and achieves similar performance to OpenHands on SWE-Bench. The paper also individually analyzes each component of the agent.

**Claims And Evidence:**

The paper's main claims are:
- We develop a code embedding model specialized in retrieving relevant code chunks to a given bug, achieving state-of-the-art file retrieval accuracies on SWE-bench: this is supported by Table 2, however additional baselines could be considered.
- We propose a diverse solution generation method by leveraging different contextual information and varied prompt formats, significantly enhancing sample efficiency: I was unsure where the sample efficiency results could be found.
- We design a localization agent that integrates advanced programming tools and leverages an ensemble of LLMs to deliver more precise and granular issue localization. Experimental results demonstrate that CORTEXA outperforms Agentless [...] while being more cost-effective: CORTEXA does outperform Agentless, but this result does not mention other baselines like OpenHands.

**Essential References Not Discussed:**

N/A

**Experimental Designs Or Analyses:**

- Evaluation is only limited to SWE-Bench. In contrast, Wang et al. 2024a evaluates on 15 benchmarks. I raise this point particularly because the performance on SWE-Bench alone is not that different from prior work, so a comparison on additional benchmarks could help distinguish the performance of CORTEXA. An additional point is that because the localization model is trained on SWE-Bench data, it would be interesting to evaluate on more diverse datasets.
- Additional experiments ablating the performance of the 3 components identified as key contributions would be helpful. Which ones affect CORTEXA performance the most? While I appreciate the individual evaluation of each component, I believe the broader community would be more interested in understanding which component to focus on when building new software agents.
- In Table 2, why do you only consider GPT-4o for Prompt? Claude would be a much stronger baseline to compare against.

**Methods And Evaluation Criteria:**

- Details on how the localization and repair stages interact with each other are unclear.
- Why did the authors select this particular model (NV-Embed-QA)? Additional justification would be helpful.
- The details about the localization models are somewhat short and not expanded on in the Appendix. For example, the authors mentions "The final dataset contains approximately 534k pairs of query-positive documents" but do not provide details about the specific distribution.

**Other Comments Or Suggestions:**

N/A

**Other Strengths And Weaknesses:**

N/A

**Questions For Authors:**

Could you please address the aforementioned questions about CORTEXA methodology and evaluation?

**Relation To Broader Scientific Literature:**

The various components of CORTEXA aim to address various pain points that have been discussed in the broader literature regarding software agents. Section 4.3-4.5 provide interesting insight into how to design these different agent components.

**Theoretical Claims:**

N/A

---

> ### Author Rebuttal · Authors · 2025-04-01
>
> Thank you for taking the time to review our work and for offering constructive feedback to help enhance our paper. Regarding the comparison to the SOTA works, please refer to our response to Reviewer WZSG. Additionally, our sample efficiency is demonstrated by achieving higher resolution rates than Agentless with only 9 patches, compared to Agentless's 40 patches.
>
> **Importance of different components:** One of the main messages of our work is that improved localization of the issue is key to increasing resolution. Our initial experiments demonstrated that when GPT-4-turbo was given the ground truth file plus additional relevant files to fill the context window, it solved only 4 Lite set instances with 1 greedy patch. Providing only the oracle file increased this to 53 instances, and narrowing to ground truth functions/classes reached 72 instances. The dramatic increase in instance resolution from narrowing the context suggests that models are severely impacted by superfluous context and can benefit from a more localized edit location. This observation is the reason why we focused our work on better localization.
>
> To verify if this phenomenon holds in our final end-to-end results, we ran new experiments by generating 9 patches with the results of the file localization vs the entity localization, as shown in response to Reviewer sboj. Using the entities retrieved by the localization agent (LA) as context, we get to pass@1 of 245, while the pass@1 with File retrieval results is 223. This is consistent with our early observations.
>
> Our second key contribution concerns generating diverse candidate patches. We induce diversity by using the variety of different contexts obtained throughout the localization process, in addition to requesting different edit formats as outputs from the language models. Using the standard temperature sampling with the LA results and search/replace edit format, we obtain a pass@9 of 270 and pass@1 of 237 after the filtering steps mentioned in Section 3.5. Using our approach to maximize patch diversity we increase the pass@9 to 307 and the pass@1 to 263. A further advantage of this patch diversity method is that it is straightforward to adopt and makes more effective use of the existing sampling budget by leveraging the strengths of different contexts. We believe this technique has been overlooked by previous works and could easily be adopted independent of Cortexa.
>
> As demonstrated in Table 2, our embedding model significantly influences file localization. To further evaluate its effect on entity localization, we performed new experiments on the Verified set by running the entity localization step with file retrieval results from the base model. The localization agent takes a list of files from the file localization step as the starting point and navigates the repository as defined in Section 3.3. Table below indicates that starting with more accurate relevant files enhances the agent’s success in identifying oracle entities.
> | Embedding Model  | Entity Precision | Entity Recall |
> |---|--|--|
> | NV-Embed-QA    | 27.62%          | 65.03%       |
> | Cortexa        | 36.22%          | 68.09%       |
>
> **Localization model:** We chose NV-Embed-QA as our pretrained model as it is the strongest commercially-available text embedding model in our tests. Its commercial availability aligns with our goal of releasing Cortexa and its associated retrieval model. Our dataset includes 290k query-positive pairs from 19k issues across 35 non-test repositories with positive passages from oracle files in the golden patch. Additionally, we generated 125k coding question-answer pairs using DeepSeek-v2.5. We also included data from APPS, CoSQA, Text2SQL, CodeTransOcean, and StackoverflowQA. We will add a detailed table of all data to the paper. While we acknowledge the potential of the embedding model for broader coding tasks, its purpose here was to help with retrieving relevant files to an issue description. A more comprehensive analysis of the embedding model on diverse datasets falls out of the scope of this paper and will be addressed in a future work.
>
> **Claude for Table 2:** We agree that several strong LLMs exist that may surpass GPT-4o here such as Claude, however we use Agentless as our baseline and thus wanted to facilitate comparisons by keeping the same LLM. Another important motivation in our choice of embedding models over prompt-based approaches is because many LLMs including GPT-4o and Claude have been trained on these exact repositories, allowing them to identify relevant files by name alone. By contrast, our embedding model does not suffer from this same test leakage and thus ensures that the gains cannot be attributed to memorization from a specific LLM. Furthermore, reading the full contents of the files with these LLMs is prohibitively costly, as shown in CodeMonkeys (Ehlrich et al. 2025), whereas our embedding model achieves equal or better accuracy using only a fraction of the cost.

---

### Official Review · Reviewer_WZSG · 2025-03-20

**Overall Recommendation:** 3

**Summary:**

The work proposes an agentic system around LLM to solve GitHub issues (Swe-bench tasks).
They mainly proposed a code embedding model used for file retrieval and built a localization, diverse patch generation and filtering mechanism around it.
Finally, they demonstrated good performance of Cortexa while being cost-effective.

**Claims And Evidence:**

yeah, the claims are clear. The evidence presented is somewhat okay but can be more convincing by reporting confidence intervals.

**Essential References Not Discussed:**

None

**Experimental Designs Or Analyses:**

Yes.
I think experiment design (or dataset) looks alright for this work. They evaluated on SWEbench which consists of real world github issues.

**Methods And Evaluation Criteria:**

The method and evaluation criteria (resolve rate, precision and recall) seem reasonable.
However they reported one number but the LLM's are quite stochastic and ideally should be run multiple times to report a confidence interval.

**Other Comments Or Suggestions:**

38: Identify what?

Why does localization via ensemble increase precision? It should reduce precision due to the stochasticity of different LLMs combined together.

Can you comment on why the performance of open hands is better than cortexa?

**Other Strengths And Weaknesses:**

Strengths:
- Paper is clearly written and easy to understand
- Novel code embedding model and its utilization to resolve the GitHub issue by using it within agentic systems

Weakness:
- Performance is weaker as compared to other SOTA methods
- Doesn't report confidence interval, so currently uncertain if we can use the results to derive conclusions

**Questions For Authors:**

see above

**Relation To Broader Scientific Literature:**

There is already some work in code agentic systems focusing on code retrieval, AST graphs, patch selection, etc.
This work shows the benefits of a fine-tuned retrieval model used in loop with coding agents to solve the github issues.
Also, there are works which achieve more than 60% in pass rate and the work doesn't connect or comment on them.

**Theoretical Claims:**

Experimental paper --> no theoretical claims.

---

> ### Author Rebuttal · Authors · 2025-04-01
>
> Thank you very much for your valuable feedback. We appreciate your thoughtful comments.
>
> **Comparison to SOTA:** In response to your comment on Cortexa's performance compared to other works, we would like to emphasize that our primary goal was to identify ways to increase the efficiency of coding agents. We observed that while free-form agentic flows are effective here, their vast pool of actions often leads to numerous costly iterations. Similar to Agentless, we aim to develop a more structured approach by identifying and improving key stages crucial for the final task, ultimately increasing efficiency and performance simultaneously. This approach also simplifies debugging and human intervention.
>
> For instance, while OpenHands achieves a 26% resolve rate on SWE-bench Lite at `$1.1/instance` (with the cost for other submissions unreported), Cortexa achieves a 42% resolve rate at `$0.51/instance`. We recognize that Cortexa's performance can be further improved by refining the selection process and/or scaling the number of generated patches. These optimizations are orthogonal to our approach and are left as a future work. Currently, the majority voting in the patch selection process is suboptimal, especially since we diversify the solutions generated, and a correct solution may not appear multiple times. Indeed, we observe that Cortexa’s performance can be increased by adopting a better patch selection approach - we’ve switched from a majority vote to using an LLM for selection, and pass@1 on the Lite set increases to 42.67% and on the Verified set to 54.8%. Additionally, another approach that brings our results in line and beyond most open approaches is scaling the number of generated patches. For instance, increasing it to 32 (16 by Claude-3.5-Sonnet and 16 by DeepSeek-V3) increases the performance on the Verified set to 58.8%
>
> **Confidence interval:** We used a temperature setting of 0 for all LLM calls, except for 4 out of 9 generated patches during the repair stage. Each of these 4 patches utilized different combinations of context and edit formats. To address the concern about stochasticity of LLM calls, we generated two additional patches for each combination (temperature sampling), resulting in 81 variations of the final 9 patches. The table below shows the peak performance statistics of these 9 patches on the Verified set. As mentioned in Appendix A.1, the patches used in the paper achieved a peak performance of 307 instances (61.4%) which falls within the confidence interval. We will include these results in the paper.
>
> | Target                          | Min  | Max  | Mean   | Std  |
> |---------------------------------|------|------|--------|------|
> | Peak perf (pass@9)     | 299  | 311  | 304.52 | 2.71 |
> | #valid patches                  | 7.75 | 7.88 | 7.79   | 0.02 |
>
> **Ensemble increases precision:**
> Indeed, this is a surprising effect we observe. In Table 4 we reported precision/recall averaged across instances. To illustrate why ensembling is helpful consider the following example retrieval task over 5 documents (A, B, C, D, E) and two queries and relevant documents {Q1, (A,B)} and {Q2, (C,D)}. Say method 1 returns (A, C) for Q1, and (A, B) for Q2; method 2 returns (C, D) for Q1, and (B, C) for Q2. Method 1’s mean precision is (1/2 + 0/2) / 2 = 1/4. Method 2’s mean precision is (0/2 + 1/2) / 2 = 1/4. The union for both methods results in (A, C, D) for Q1 and (A, B, C) for Q2. The ensemble mean precision is (1/3 + 1/3) / 2 = 1/3 greater than either methods’ precision (1/4). For each question instance, the union ensemble has a lower precision (1/3) than the highest individual precision (1/2). This phenomenon explains why our ensemble method benefits from models’ strengths in solving different problem subsets.

---

### Decision · Program_Chairs · 2025-05-01

**Decision:**

Accept (poster)

**Comment:**

This paper introduces CORTEXA, an agentic system designed to improve Large Language Models on software engineering tasks by enhancing localization and solution diversity. Reviewers acknowledged the clarity of the presentation (WZSG), the novelty of combining a fine-tuned embedding model with an agentic approach for localization (WZSG, sboJ), and the interesting direction of improving efficiency via diverse patch generation (sboJ, KBKm). However, initial concerns were raised regarding the performance compared to state-of-the-art methods (WZSG, KBKm), the lack of statistical significance testing or confidence intervals (WZSG, sboJ), the limited evaluation scope and missing baselines (KBKm, sboJ), and insufficient methodological details impacting reproducibility (KBKm, sboJ).

The authors provided a detailed rebuttal including substantial new experimental results and clarifications. Concerns about performance were addressed by presenting results from scaled-up experiments and improved selection mechanisms, demonstrating significant gains over the primary baseline (Agentless) and competitiveness with other relevant methods, often at lower cost. Statistical significance was established for key claims through p-value reporting and confidence interval analysis derived from additional runs. The authors also included new ablations clarifying the impact of the localization and diversity components, added comparisons to more contemporary agentic retrieval baselines, provided further methodological details, and committed to releasing code and models, addressing reproducibility concerns.

Overall, the initial reviews highlighted valid points requiring clarification, while the extensive rebuttal and promised changes effectively mitigated the major weaknesses. The strengthened empirical evidence, particularly the demonstrated performance gains achieved through improved localization and patch diversity with a clear focus on efficiency, now supports the paper's contributions. The work presents valuable insights and techniques for developing more effective LLM-based software engineering agents, justifying its inclusion in the conference program.